# Core/Shell Pigments with Polyaniline Shell: Optical and Physical–Technical Properties

**DOI:** 10.3390/polym14102005

**Published:** 2022-05-13

**Authors:** Tatyana A. Pugacheva, Georgiy V. Malkov, Alexander A. Ilyin, Eugene A. Indeikin, Vladimir G. Kurbatov

**Affiliations:** 1Institute for Problems of Chemical Physics of the Russian Academy of Sciences (IPCP RAS), Academician Semenov Avenue 1, 142432 Chernogolovka, Russia; sinichka.71@yandex.ru (T.A.P.); gmalkov@mail.ru (G.V.M.); 2Department of Chemical Technology of Organic Coatings, Yaroslavl State Technical University, Moscow Avenue 88, 150023 Yaroslavl, Russia; 471562@mail.ru (A.A.I.); eugene.indeikin@ya.ru (E.A.I.)

**Keywords:** polyaniline, core/shell pigments, optical properties, inorganic fillers, doping

## Abstract

Core/shell pigments allow for the combination of the active anti-corrosion effect of the shell and the barrier effect of the core. This makes it possible to obtain anti-corrosion pigments, with a high—protective effect and low toxicity. Thus, the need for a comprehensive study of the properties of these pigments grows more urgent, before their application to paints and varnishes. The hiding power of core/shell pigments comes close to the one of pure polyaniline (PANi), when the PANi content in the pigment reaches 50 wt.%, with sulfuric and phosphoric acids used as dopants. This paper, also, shows that the blackness value of core/shell pigments with 10 wt.% PANi is around 35 and constant; for pure PANi, their blackness value is 40. When PANi content is 5 wt.%, kaolin-based pigment shows the lowest blackness, which happens due to a generally higher whiteness of kaolin. However, when the PANi content surpasses 10 wt.%, there seems to be no influence on the blackness of the core/shell pigments. The core/shell pigment with a 20 wt.% PANi is, optically, identical to a black-iron-oxide pigment. An increase in the PANi content of the core/shell pigment leads to an increase in the oil absorption of the samples. It was found that the dispersion process would be the most energy efficient for core/shell pigments, containing kaolin and talc as a core.

## 1. Introduction

Aniline-based dyes were first obtained in the 19th century [1]. Aniline blue, yellow, orange, black, and a wide range of other dyes were synthesized, in accordance with industry demands, by the end of the 19th century. Aniline-based-dye production led to the abandonment of scarce and expensive natural raw materials (indigo, madder, cochineal, Tyrian purple). Today, the «Colour Index» international pigment and dye classifier contains more than 150 aniline-based products as well as others based on derivatives. Dyes obtained using aniline can be divided into three groups [2,3]:
Dyes made from aniline;Dyes that include other compounds, apart from aniline;Dyes based on intermediate products made from aniline.

However, aniline-based dyes possess bad substrate coverage, which does not allow them to be added to paints and varnishes as a pigment. Polyaniline (PANi), the product of aniline polymerization in the presence of various oxidizing agents, is an exception. PANi possesses poor solubility in most organic solvents, which limits its practical implementation. However, obtaining soluble PANi is still a top-level task for a lot of researchers all over the world. This fact made it possible to obtain PANi films with high conductivity and high mechanical properties [4,5,6,7]. For example, water-soluble PANi was synthesized in [4], by its sulfonation with sulfuric acid. Water-soluble PANi films can be used as protective coatings and, also, as an active layer for the detection of various substances, which significantly expands its application. A number of papers [8,9,10,11] show that samples of PANi soluble in a number of organic solvents were obtained. To make it possible, in particular, miniemulsion polymerization was applied. This form of PANi is able to dissolve, e.g., in chloroform, dimethyl sulfoxide, toluene, and xylene. It should be noted that the PANi yield was reported to be low in a number of papers. The films obtained, however, possessed high conductivity, high mechanical properties, and high thermal stability. Nevertheless, the authors succeeded in using cheap diesel fuel as a dispersion medium, to obtain a soluble product with a high yield [12].

Another way to introduce PANi into polymeric composite materials and, in particular, paints and varnishes, is to use it as a pigment.

It is known [13] that poor dispersibility is one of the significant disadvantages of organic pigments. The reason for that is primary particles possess high dispersity and strong aggregation, which, in turn, increases energy consumption during the production of paints and varnishes. Another essential flaw, typical for organic pigments, including PANi, is their high oil absorption. Due to this, the effective critical pigment volume concentration decreases. One of the possible solutions is to apply a thin-shell PANi application to an inert medium surface.

A variety of publications reported the high anticorrosive performance of such structures [14,15,16,17,18,19,20]. Along with fine protective properties, PANi-containing shell structures possess increased electrical conductivity [21,22,23,24,25,26], which can help to obtain antistatic coatings [27,28,29,30,31,32]. However, optical and physical–technical pigment properties play a significant role in the development of the composition of paints and varnishes. Core/shell pigment optical properties are set out by PANi content in the composition, as well as by the form of PANi—doped or undoped. Therefore, it is necessary to evaluate the effect of the content and form of PANi on the optical and physical–technical properties of core/shell pigments.

## 2. Materials and Methods

In total, 0.28 g of H_2_SO_4_ and 0.25 g of aniline (Sigma-Aldrich, St. Louis, MO, USA) were dissolved into 50 g of water and stirred well with a magnetic stirrer, until the resulting aniline salt was completely dissolved. Then, 5 g of core pigment was added into the reaction mixture. After that, 50 mL of aqueous solution of ammonium persulfate (0.75 g of ammonium persulfate (Sigma-Aldrich)/49.25 g of water) was added, simultaneously, to the suspension, obtained under intensive stirring. The reaction mixture was stirred for 1 h and, then, left for 24 h. The resulted suspension was filtered and washed, first with 250 mL of acid solution used in the synthesis (1 mol/L) and, then, with acetone, until a clear filtrate was obtained. The core pigment obtained was dried at 60 °C for 12 h.

Inorganic fillers, such as kaolin (KaMin™ HG, KaMin LLC, Lawrenceville, GA, USA), talc (Talc-8, Ashirwad Minerals and Marbles), bentonite (BENTONE SD-1, Elementis Specialties, London, UK), and muscovite (Micro-Mica W1, New Enterprise Stone & Lime Co., Inc., New Enterprise, PA, USA), which could serve as new solid-phase nuclei during polymer formation, were also presented in the medium. The main properties of the used inorganic fillers are shown in Table 1. Sulfuric, phosphoric, and hydrochloric acids (all acids were purchased from Sigma-Aldrich and used as received or purified according to standard procedures) were used as PANi dopants. The PANi content in the core/shell pigment varied from 0 to 50 wt.%.

Optical micrographs were taken using a Biolam S12 (LOMO, Saint Petersburg, Russia) optical microscope. SEM micrographs were taken using a Zeiss SUPRA 40 (Carl Zeiss, Oberkochen, Germany) electron microscope. Fourier-transform infrared spectroscopy (FTIR) analysis of PANi salt powder and core/shell pigments was carried out in the range of 375–4000 cm^−1^ using BRUKER ALPHA (Bruker Optics, Billerica, MA, USA). Oil absorption of the samples was determined using [33]. Density was measured by pycnometry [34]. The hiding power of the core/shell pigments obtained was determined visually [35]. Color properties of core/shell pigments were obtained using a spectrophotometer Ci4200 (X-rite, Kentwood, MI, USA). The pigment blackness was measured according to [36,37].

The coloring ability of core pigments with PANi coating was determined as follows. The paste with a core/shell pigment to titanium dioxide ratio of 1:3 was prepared by grinding on the plate using a chime and spatula, until it became homogeneous. Then, this paint was used to make paint touches, with their reflection spectra being detected on a spectrophotometer (X-rite Ci4200). The *p* minimum value was determined for the test and control samples, in the short-wave area of the spectrum. The two-constant Kubelka–Munk theory function was calculated using the formula [34],
(1)$$F\equiv K/S={\left(1-\rho \right)}^{2}/2\rho$$
where *ρ* is the minimum reflection index value for the determined wavelength.

Black-iron-oxide pigment was used as the reference sample. Coloring ability *I*_col_ (%) was calculated by:(2)$$I_{\mathrm{col}}=\frac{F_{\mathrm{test}}}{F_{\mathrm{cnt}}}\cdot 100$$
where *F*_test_ and *F*_cnt_ are Kubelka–Munk functions of test and control (iron oxide black) samples, respectively. 

To determine pigment dispersibility, the pigment paste made, using a specific recipe, was dispersed in a bead mill. The recipe is: water prepack 7.5 g (consists of water, ethylene glycol, sodium tripolyphosphate, hydroxyethyl cellulose), pigment 5 g (0.83 g of PANi-coated core/shell pigment and 4.17 g of titanium dioxide), and the pigment paste to beads ratio is 1:1. Approximately 0.5 mL of the paste were sampled after 5, 15, 25, 35, 50, and 65 min. A piece of thick paper was covered with the obtained paste, using a brush. The painting-touch size for spectrum detection was 50 mm × 50 mm. The reflection index of the hardened sample was registered on the spectrophotometer Ci4200 (X-rite, Kentwood, MI, USA), at the minimum of sample reflectivity. Using the obtained indices, Kubelka–Munk function values were calculated, and, then, *t/F* vs. *t* graph was plotted. After that, *k*_d_ and *F_∞_* were calculated using the least squares method. *k*_d_ was determined as a reciprocal of the length of the Y-axis line, cut with the obtained graph, while *F_∞_* was determined as a reciprocal of the slope [34].

Dispersion resistance value was calculated by the formula [34]:(3)$$t_{0.5}=\frac{F_{\infty }}{k_{d}}$$
where *F_∞_* is the Kubelka–Munk function critical value, and *k*_d_ is dispersion rate constant.

## 3. Results and Discussion

Based on the optical microscopy data, it can be seen that core/shell pigment is formed during the course of the synthesis (Figure 1, Figure 2 and Figure 3). Individual PANi particles and colorless lamellar talc particles can be seen in the micrographs of the mechanical mixture of talc and PANi (Figure 4).

The pictures obtained using scanning electron microscopy, also, confirm the formation of core/shell pigments (Figure 5).

The FTIR spectra of different core/shell pigments and PANi samples are shown in Appendix A. The absorption bands at 1478 cm^−1^ and 1560 cm^−1^ are assigned to benzenoid and quinoid ring vibration, respectively [38]. The absorption band characteristic of the conducting protonated form is found at about 1245 cm^−1^. The absorption band at 1139 cm^−1^ is due to the in-plane bending vibration of C–H [39], while at 1305 cm^−1^ it is because of C–N stretching of the benzenoid ring of PANi [40]. The aromatic ring and out-of-plane C–H deformation vibrations manifest themselves in the region of 900–700 cm^−1^. The FTIR spectra of core/shell pigments show a blue shift of the main absorption bands, compared to PANi. Besides, from absorption bands caused by PANi presence, there are a number of bands in the spectra of core pigments related to the core used in the study. For kaoline: 3692 cm^−1^, 3666 cm^−1^, and 3653 cm^−1^—O-H stretching vibrations of inner surface hydroxyl bands, 3620 cm^−1^—O-H stretching vibrations of inner hydroxyl bands, 1115 cm^−1^—Si-O stretching vibrations, 1032 cm^−1^ and 1010 cm^−1^—in-plane Si-O stretching vibrations, 914 cm^−1^—O-H deformation of inner hydroxyl bands, 787 cm^−1^—Si-O stretching vibrations, 753 cm^−1^—Si-O stretching vibrations, 541 cm^−1^—Al-O-Si deformation vibrations, and 471 cm^−1^—Si-O-Si deformation vibrations (Appendix A). For talc: 3673 cm^−1^—vibrations of hydroxyl groups linked to Mg-OH, 1040 cm^−1^ and 476 cm^−1^—siloxane group (Si-O-Si) stretching vibrational bands, and 677 cm^−1^—corresponds to vibrations Si-O-Mg bond (Appendix A). For bentonite: 3695 cm^−1^—assigned to Al-OH-Al in the mineral, 3620 cm^−1^—O-H stretching vibrations of inner hydroxyl bands, 1035 cm^−1^—assigned to the degenerate Si-O stretching in-plane vibration, and 913 cm ^−1^ is assigned to OH deformation mode of Al-Al-OH or Al-OH-Al, (Appendix A). For muscovite: 3695 cm^−1^—O-H stretching vibrations of inner surface hydroxyl bands, 3625 cm^−1^—O-H stretching vibrations of inner hydroxyl bands, 1115 cm^−1^—Si-O stretching vibrations, 1020 cm^−1^ and 996 cm^−1^—in-plane Si-O stretching vibrations, 826 cm^−1^—Al-O stretching (Al in tetrahedral sheet), 747 cm^−1^—Al-O-Al stretching (Al in tetrahedral sheet), 532 cm^−1^—Al-O-Si deformation vibrations, and 478 cm^−1^ and 416 cm^−1^—Si-O-Si deformation vibrations (Appendix A).

Reflection spectrum has a maximum in the 500–550 nm range for all PANi-shell core/shell pigments (Figure 6). A change in PANi content in core/shell pigment affects the reflection index only and does not lead to a shift in the absorption maximum (Figure 7a). Replacing doped PANi with undoped shifts the absorption maximum to the shortwave area of 420–450 nm (Figure 7b).

A change in the shell thickness, as well as the nature of PANi surface, leads to changes in color characteristics. The study established the influence of shell thickness and the nature of PANi surface on color properties. The graphs show that an increase in the doped PANi content raises the **b*** coordinate and lowers the **a*** coordinate. The core/shell pigment-color tone moves from a green area to a green-yellow area. The **b*** coordinate growth, with an increase in the PANi content for core/shell pigments, may be due to a large number of yellow-brown-colored aniline oligomers (Figure 8a–c). An inverse relationship is observed for core/shell pigment with undoped PANi coating: the **a*** coordinate rises and the **b*** coordinate declines with an increase in the PANi content (Figure 3d). However, this happens because the core/shell pigment covered with undoped PANi is blue, and the mixture turns greener due to the color subtractivity. The angle of the color vector projection shows it clearly.

Hiding power is another important optical property of pigment. This property, for PANi-coated core/shell pigments, depends on the polymer content in the pigment only and has no relation to the doping acid used in the synthesis. Core/shell pigment hiding power grows with an increase in PANi content (Figure 9). It should be noted that the hiding power is comparable to that of pure PANi, even at a 50 wt.% of core/shell pigment content, with a shell of phosphoric-acid-doped PANi.

The same relation can be seen for the core/shell pigment-coloring strength (Figure 10)—it rises along with the increase in PANi content. 

The obtained core/shell pigments possess low color purity, so their application as green pigments is objectionable. However, high blackness may let it be applied as black pigments, since one of their important properties is blackness. Figure 11 shows that the blackness of core/shell pigments increases with the growth of PANi content. Moreover, the nature of the core used affects the blackness of core/shell pigments, only at low PANi content (~10 wt.%). So, a further increase in PANi content results in pigments that possess approximately equal blackness, regardless of the used core. The blackness of all core/shell pigments is slightly lower compared to PANi. Doping acid affects core/shell pigment blackness poorly, regardless of the PANi content. Figure 12 shows the core/shell pigment blackness vs. PANi content graph.

Despite the high absorption index, black pigments are, usually, diverse in their shade of black. Growth of PANi content in core/shell pigment causes a brown tint for almost all samples, due to the brown aniline oligomers in its content. Their mass is not high enough to count PANi in, but they have a high molecular weight, enough to remain undissolved in acetone while washing off the core/shell pigments. Table 2 shows the comparison of core/shell pigments, with the coating of PANi and common black pigments.

It is clear from the table that the core/shell pigments obtained with a PANi shell (20 wt% and more) are similar to iron-black pigment, and an increase in PANi content results in an excess of all optical properties.

Anticorrosive features, also, stand out as an advantage of core/shell pigments. Being coated, they can possess a barrier effect, due to a lamellar and/or flaky-core shape. PANi presence, also, provides an active anticorrosive defense. 

It is essential to know the physical–technical properties of the pigments, besides their optical properties, when it comes to developing polymer-material compositions. The type of doping acid, the PANi content in the core/shell pigment composition, and the core used during synthesis can influence the properties. One of the important indices defining the pigment content in filled material is oil absorption. The presence of a polymer shell increases the surface of the core/shell pigment. Figure 13 shows the oil absorption vs. PANi content in the core/shell pigment. Oil absorption for PANi is much higher, in comparison to the one for initial fillers. This is associated with a larger content of PANi, which has a higher oil absorption compared to the filler used as a core. This is associated with a larger specific surface area of PANi, and, therefore, a large amount of oil is required for wetting the powder (specific surface area core/shell pigment (20% of doped PANi, muscovite-based, H_2_SO_4_ as a dopant, 3.1 m^2^/g); specific surface area core/shell pigment (50% of doped PANi, muscovite-based, H_2_SO_4_ as a dopant, 9.1 m^2^/g)). Due to a rise in core/shell pigment specific surface area, an increase in PANi content leads to an increase in oil absorption. This holds for all doping agents used. However, for hydrochloric acid as a dopant, the function is a bit more complicated than that for sulfuric and phosphoric acids. PANi shell doping with hydrochloric acid results in the rise of oil absorption to a polymer content of 10 wt.%, and, then, it remains almost immobile up to 50 wt.%. That functional behavior may indicate a complete uniform overlap of the PANi core surface. A similar, anomalous behavior is observed for all core pigments, where hydrochloric acid was used as the PANi doping agent (Figure 14).

For some samples, the extreme type of oil absorption vs. PANi content graph may be attributable to the fact that the maximum oil absorption is observed when the core surface is completely covered by the PANi shell. A further decrease in the function, with an increase in PANi content, occurs due to “secondary fouling” [41,42], which reduces the specific surface area of a core pigment particle.

The density of core/shell pigments heavily depends on the ratio of the inorganic core and organic shell. Furthermore, doping acid can influence the density of obtained pigments, since the producing PANi density changes in accordance with it. Figure 15 shows the effect of PANi content in the core/shell pigment on its density. 

It is clear that the growth of PANi content in core/shell pigment corresponds with density reduction. The reason for that is the lower density of PANi over the starting filler. Therefore, an increase in PANi content leads to a decrease in core/shell pigment density. PANI doping agent, used marginally, affects the core/shell pigment density. The same trends are observed for using other substrates as a core material.

Dispersibility is the other important pigment performance, for making filled materials. Table 3 shows the results of studying the core pigment dispersibility. 

It can be seen that, regardless of PANi content, the dispersion rate constant is the highest for all core/shell pigments using hydrochloric acid as a dopant for PANi. The time to reach half of the Kubelka–Munk function limit value lies within 10 min, for most of the core pigments. 

That characteristic can be arranged in the following order (Table 4), depending on the core used: muscovite > bentonite > talc > kaolin. So, kaolin- and talc-based core/shell pigment dispersing is the lower energy-intensive process. 

## 4. Conclusions

It was shown that the hiding power of the obtained core/shell pigment samples increases with the increase in PANi content, regardless of the doping agent used. Their coloring strength, also, grows as the PANi content increases. Core/shell pigments with PANi shell doped with phosphoric acid possess the best coloring strength. 

In addition, it was demonstrated that the core/shell pigment color shade shifts from a green area to a green-yellow area, with the PANi content increase. This was due to the high content of aniline oligomers. The core/shell pigment blackness depends directly on the PANi content. 

It was established that a growth in PANi content leads to an increase in core/shell pigment oil absorption. This results from pigment-specific surface area expansion. It was found that core/shell pigment density reduces with the rise of PANi content, owing to the lower density of the conducting polymer particles.

The core/shell pigment dispersing was studied. It was shown that kaolin- and talc-based core/shell pigments, with PANi shell doped with sulfuric and hydrochloric acids, are well-dispersed.

It was established that core/shell pigments can be used as black pigments in the composition of paints and varnishes.

## Figures and Tables

**Figure 1 polymers-14-02005-f001:**
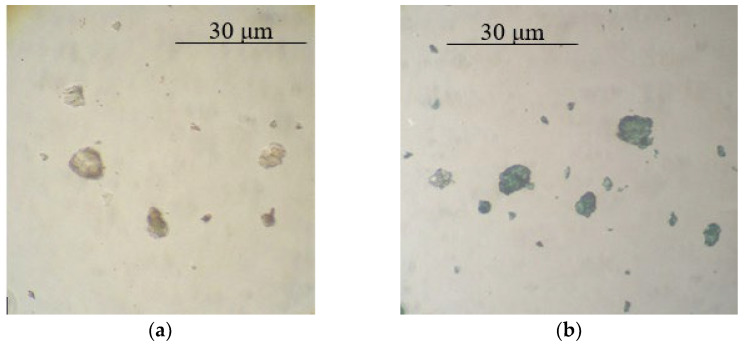
Micrographs of bentonite (**a**) and core/shell pigment with a bentonite core (**b**) (20 wt.% of PANi, sulfuric acid used as a dopant).

**Figure 2 polymers-14-02005-f002:**
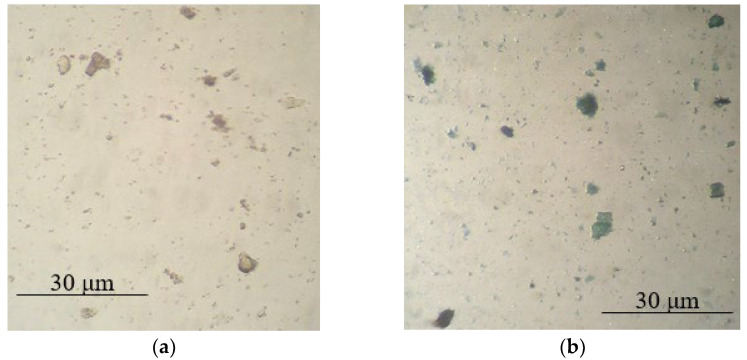
Micrographs of kaolin (**a**) and core/shell pigment with a kaolin core (**b**) (20 wt.% of PANi, sulfuric acid used as a dopant).

**Figure 3 polymers-14-02005-f003:**
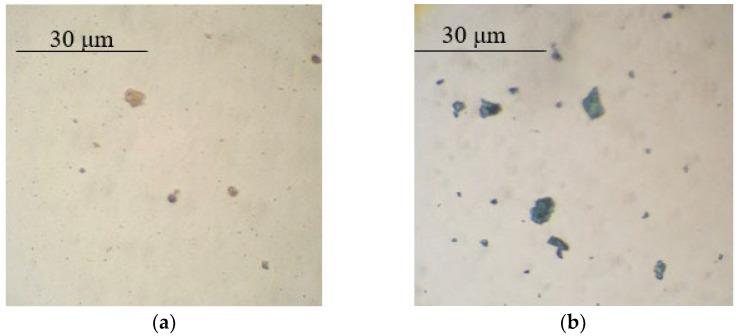
Micrographs of mica (**a**) and core/shell pigment with a mica core (**b**) (20 wt.% of PANi, sulfuric acid used as a dopant).

**Figure 4 polymers-14-02005-f004:**
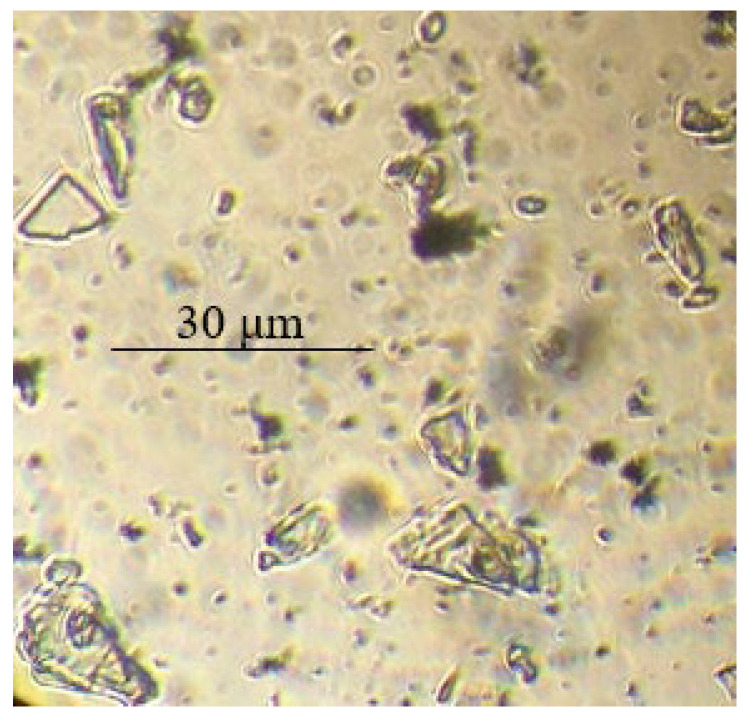
Micrographs of the mechanical mixture of H_2_SO_4_-doped PANi (20 wt.%) and talc (80 wt.%).

**Figure 5 polymers-14-02005-f005:**
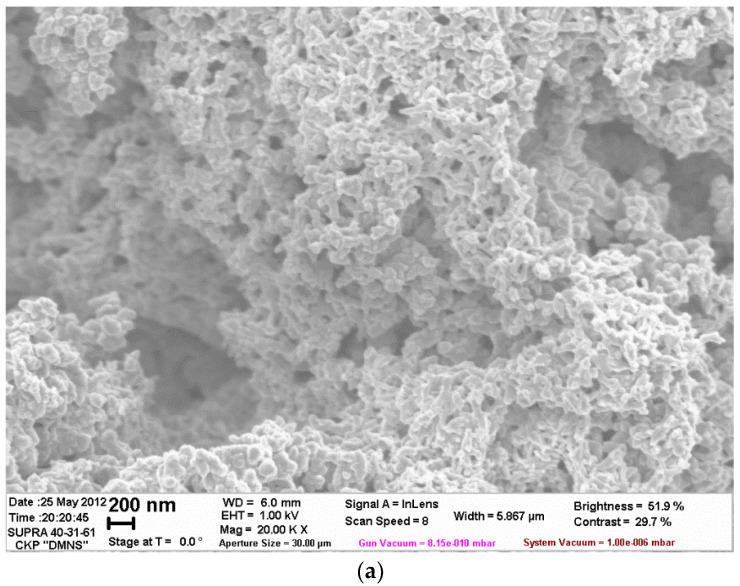
SEM image of core/shell pigment with a talc core and H_2_SO_4_-doped PANi shell (**a**)—20,000× magnification, (**b**)—3000× magnification).

**Figure 6 polymers-14-02005-f006:**
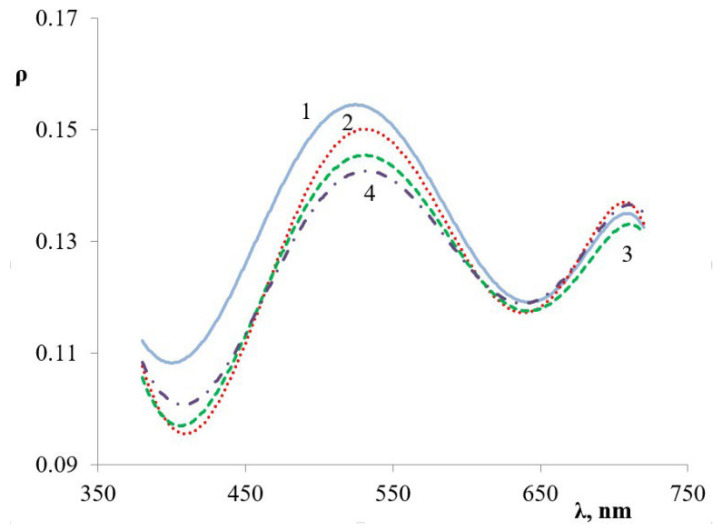
Reflection spectra of core/shell pigment samples with doped PANi coating. The core of the pigment: (1) kaolin, (2) talc, (3) bentonite, (4) muscovite.

**Figure 7 polymers-14-02005-f007:**
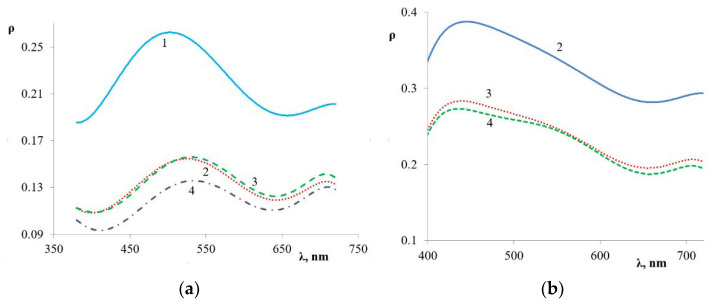
Reflection spectra of core/shell pigment samples with (**a**) doped and (**b**) undoped PANi coating. PANi content in the core/shell pigment (wt.%): 1—5; 2—20; 3—33; 4—50.

**Figure 8 polymers-14-02005-f008:**
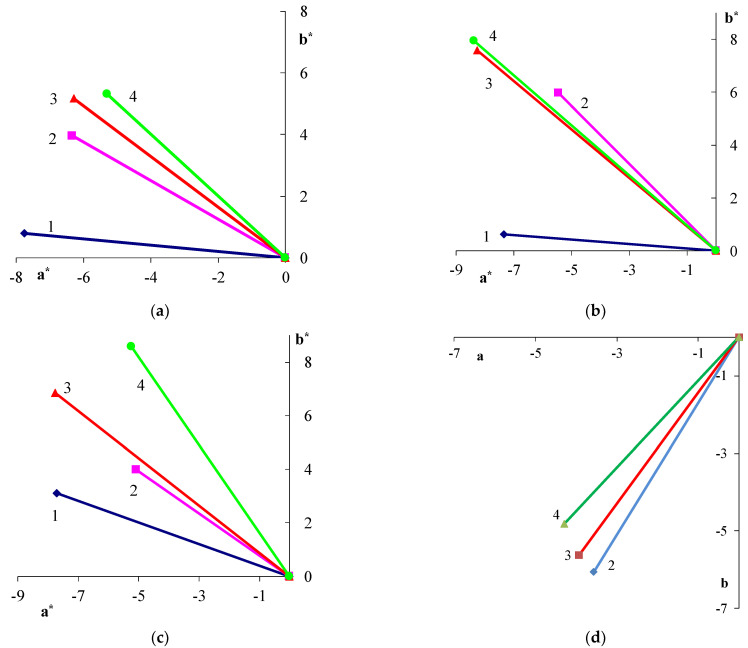
Color vector projection of core/shell pigment based on talc and doping acid: (**a**) H_2_SO_4_; (**b**) HCl; (**c**) H_3_PO_4_; (**d**) undoped PANi on the CIEL*a*b* color plane. PANi content in the core/shell pigment (wt.%): (1)—5; (2)—20; (3)—33; (4)—50.

**Figure 9 polymers-14-02005-f009:**
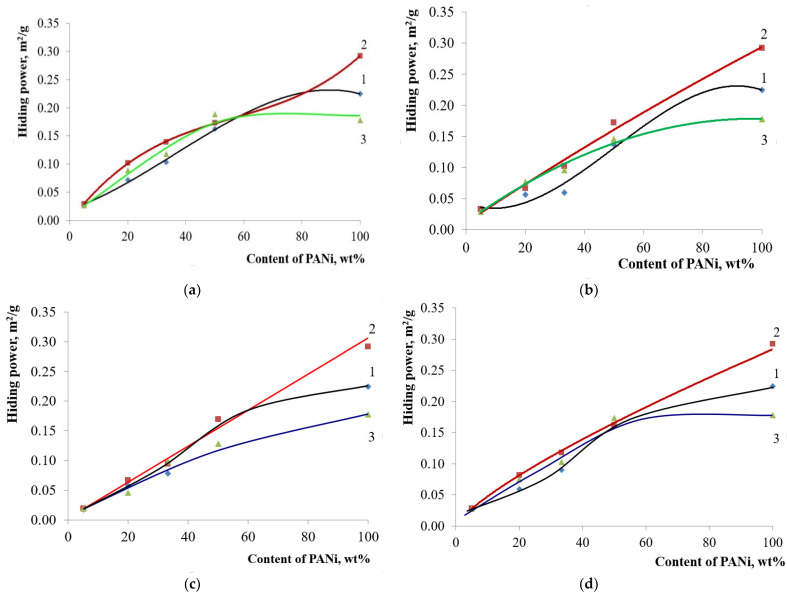
Hiding power of core/shell pigments with a various base vs. the content of PANi doped with various acids: (**a**) kaolin-based core/shell pigment; (**b**) bentonite-based core/shell pigment; (**c**) talc-based core/shell pigment; (**d**) muscovite-based core/shell pigment. Doping acid: (1)—H_2_SO_4_; (2)—HCl; (3)—H_3_PO_4_.

**Figure 10 polymers-14-02005-f010:**
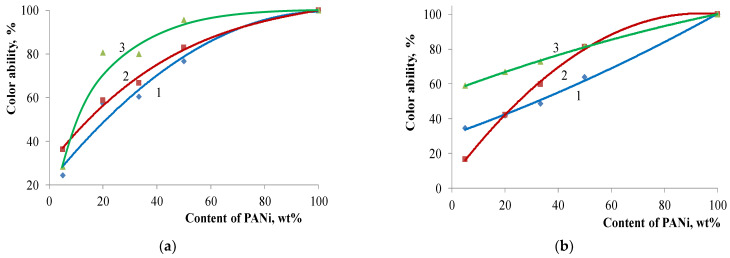
Coloring strength of core pigments with a various base vs. the content of PANi doped with various acids: (**a**) kaolin-based core/shell pigment; (**b**) bentonite-based core/shell pigment; (**c**) talc-based core/shell pigment; (**d**) muscovite-based core/shell pigment. Doping acid: (1) H_2_SO_4_; (2) HCl; (3) H_3_PO_4_.

**Figure 11 polymers-14-02005-f011:**
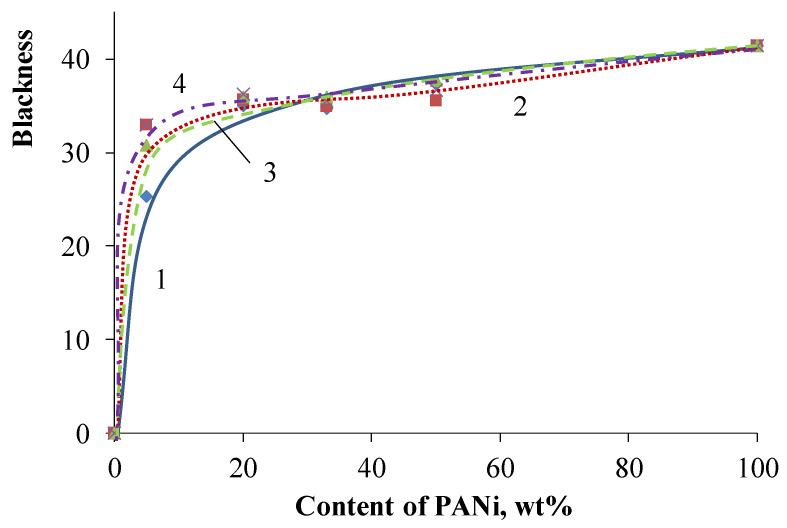
Core/shell pigment blackness vs. PANi content with various cores. The core of the pigment: (1) kaolin; (2) talc; (3) bentonite; (4) muscovite.

**Figure 12 polymers-14-02005-f012:**
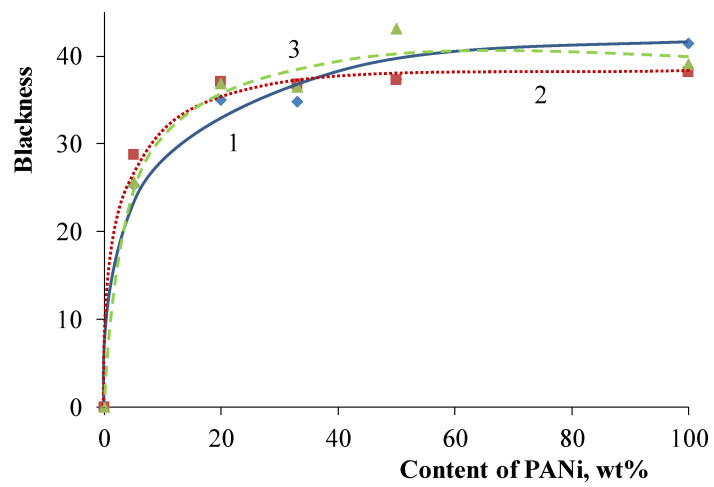
Core/shell pigment blackness vs. PANi content with various doping acids. Doping acid: (1) H_2_SO_4_; (2) HCl; (3) H_3_PO_4_.

**Figure 13 polymers-14-02005-f013:**
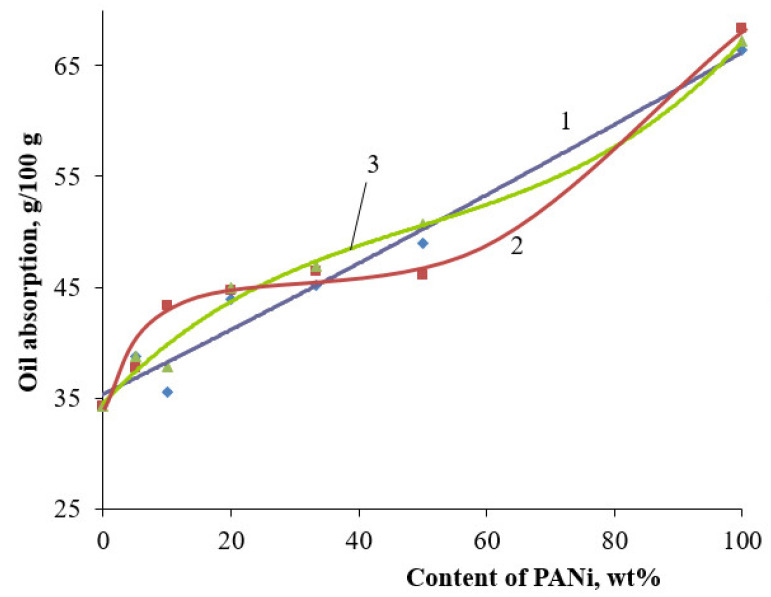
Dependence of oil absorption of core/shell pigments with a talcum core, from the content of PANi doped with various acids. Doping acid: (1)—H_2_SO_4_; (2)—HCl; (3)—H_3_PO_4_.

**Figure 14 polymers-14-02005-f014:**
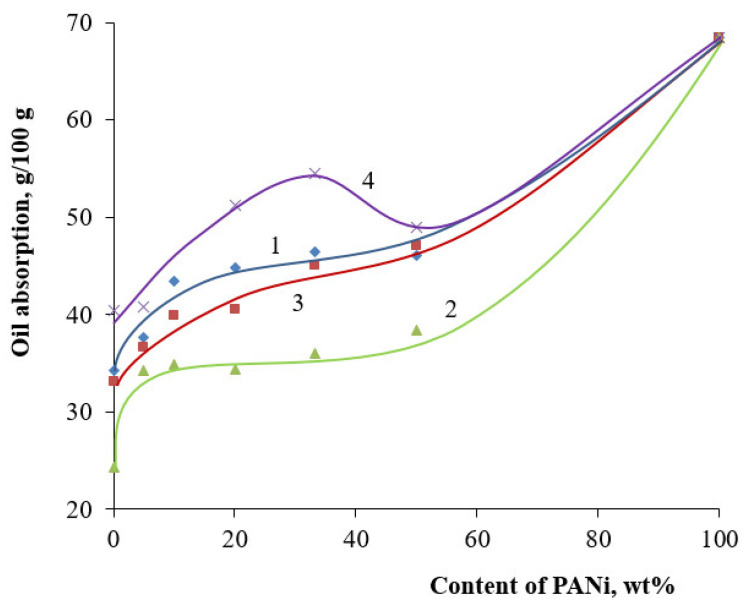
Core/shell pigment oil absorption vs. PANi content doped with hydrochloric acid. Core: (1)—kaolin, (2)—talc, (3)—bentonite, (4)—muscovite.

**Figure 15 polymers-14-02005-f015:**
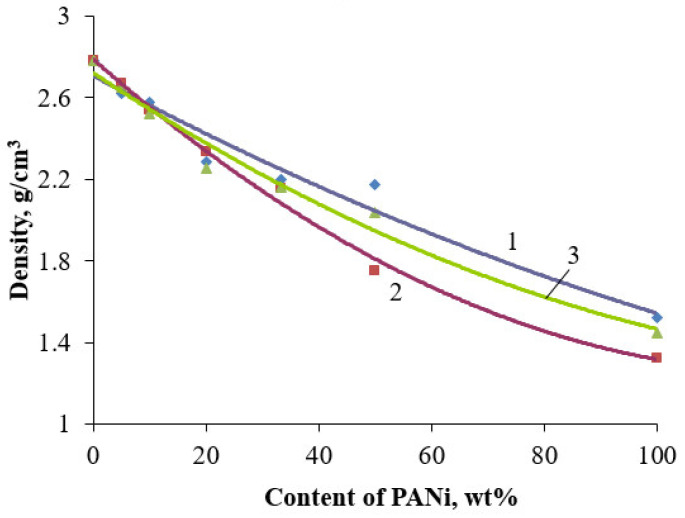
Dependence of the density of core/shell pigments on the content of PANi doped with various acids (core/shell pigment based on talc). Doping acid: (1)—H_2_SO_4_; (2)—HCl; (3)—H_3_PO_4_.

**Table 1 polymers-14-02005-t001:** Properties of used fillers.

	Filler	Kaolin	Bentonite	Talc	Muscovite
Characteristic					
Refractive index		1.6	-	1.58	1.59
Density, kg/m^3^		2600	2450	2950	2900
pH of water extract		5–8	4–6	8–10	3.6–4.3
Oil absorption, g/100 g		34.3	33.0	24.5	40.5

**Table 2 polymers-14-02005-t002:** Black pigments comparison.

Pigment	Hiding Power, m^2^/g	Coloring Ability, %
Core/shell pigment (20% of doped PANi, kaolin-based, H_3_PO_4_ as a dopant)	0.102	106
Iron black	0.132	100
Doped PANi	0.224	131

**Table 3 polymers-14-02005-t003:** Dispersibility of core/shell pigment with PANi shell doped with various acids.

Dopant	*k* _d_	*t*_0.5_, min
5 wt.% PANi		
H_2_SO_4_	0.012	3.53
HCl	0.015	3.11
H_3_PO_4_	0.009	4.73
20 wt.% PANi		
H_2_SO_4_	0.028	2.79
HCl	0.039	3.72
H_3_PO_4_	0.008	20.3
33.3 wt.% PANi		
H_2_SO_4_	0.020	12.54
HCl	0.030	5.66
H_3_PO_4_	0.026	7.74
50 wt.% PANi		
H_2_SO_4_	0.135	1.91
HCl	0.145	2.00
H_3_PO_4_	0.027	15.21

**Table 4 polymers-14-02005-t004:** Dispersibility of core/shell pigment with PANi shell doped with sulfuric acid.

Pigment Core	*k* _d_	*t*_0.5_, min
Kaolin	0.123	1.68
Talc	0.028	2.79
Bentonite	0.037	3.30
Muscovite	0.042	3.70

## Data Availability

The data presented in this study are available in this article.

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
