# Peer review of "Core/Shell Pigments with Polyaniline Shell: Optical and Physical–Technical Properties"

_polymers, 2022, doi:10.3390/polym14102005_

Round 1
Reviewer 1 Report
In this manuscript, the authors investigated the effect of the content and form of PANI on the optical properties and physical-technical properties of core pigments, some comments should be considered as follow:
- In “Materials and Methods”, the details of the synthesis of core/shell pigments should be introduced.
- What methods were used to identity the core-shell structure of the pigment? Some characterized results such as IR, SEM, TEM should be added.
- What is the sample size of the pigment in all the figures?
- Abbreviations should be described in full name only when appeared the first time in the manuscript, there is no need to repeat it, such as “polyaniline (PANi)” in entry 37 and 45.
- The subscript format should be carefully checked, such as m2/g and H3PO4 in table 2.
Author Response
Point 1: In “Materials and Methods”, the details of the synthesis of core/shell pigments should be introduced.
Response 1: The manuscript was supplemented with a detailed description of core/shell pigment synthesis.
Point 2: What methods were used to identity the core-shell structure of the pigment? Some characterized results such as IR, SEM, TEM should be added.
Response 2: Optical microscopy, FTIR spectroscopy and SEM were used to identify the structure of core/shell pigments. The results obtained using these methods were added to the manuscript.
Point 3: What is the sample size of the pigment in all the figures?
Response 3: The size of the pigment sample on all graphs was no more than 1 µm according to the grindometer.
Point 4: Abbreviations should be described in full name only when appeared the first time in the manuscript, there is no need to repeat it, such as “polyaniline (PANi)” in entry 37 and 45.
Response 4: The changes were added to the manuscript.
Point 5: The subscript format should be carefully checked, such as m2/g and H3PO4 in table 2.
Response 5: The changes were added to the manuscript.
Reviewer 2 Report
This article describes the preparation of core/shell pigments with the use of one of the most fascinating conducting i.e. polyaniline (PANI) as shell. The core shell pigments were prepared by oxidative polymerization of aniline using HCl, H2SO4 and H3PO4 as dopants in the presence of inorganic fillers such as kaolin, talc, bentonite and muscovite. The effect of the content and form of PANI on the optical and physicotechnical properties of core pigments are presented in the form of hiding power, colourability, blackness and oil absorption. The work is interesting because optical and physicotechnical pigment properties play a significant role in the paints and varnishes composition development. However, major revision of the article is required before its suitability for publication in in the light of following comments:
Abstract
- Abstract is lacking the introductory sentence about the importance of core/shell pigments.
- The very first sentence in the abstract seems to be incomplete.
Introduction
- They write “However, aniline based dyes possess bad substrate coverage that doesn’t allow them to be added to paints and varnishes as a pigment. Polyaniline (PANi), the product of aniline polymerization in the presence of various oxidizing agents, is an exception. The facility to cover the substrate and its poor solubility in almost all dispersed phases makes it possible to use it as a pigment in polymer composite materials”
It is suggested that the authors highlight exceptional properties of PANI in the light of latest literature ( for guidance please read Colloids and Surfaces A: Physicochemical and Engineering Aspects 626 (2021) 127076, Electrochimica Acta 320 (2019) 1345, Materials 2019, 12, 1626; Nanomaterials 2020, 10, 118 and others).
Polyaniline has been researched extensively and there are reports on the synthesis of soluble PANI. Authors should discuss it in more detail (please read Materials 12, 1626 (2019), Materials 12, 1227 (2019) and Synth. Metals 162 (2012) 2259-2266.
Materials and methods
- Incorporate reference for equation 3.
Results and Discussion
- It would be more appropriate to characterize PANI shells with some more spectroscopic techniques such as FTIR.
- The better performance of doped PANI shell has been attributed to the large surface area of PANI. However, one cannot find surface area measurements in the whole manuscript. Similarly morphological characterization would add additional merits to the PANI based shells for paints and varnishes composition development.
- Is there any correlation between the electrical conductivity and physicotechnical pigment properties of PANI based shell?
References
- Out of the 30 references, 3 are from the corresponding author of this article. This reflects self-citation.
Author Response
Point 1 (Abstract): Abstract is lacking the introductory sentence about the importance of core/shell pigments.
Response 1: The changes were added to the manuscript.
Point 2 (Abstract): The very first sentence in the abstract seems to be incomplete.
Response 2: The changes were added to the manuscript.
Point 3 (Introduction): They write “However, aniline based dyes possess bad substrate coverage that doesn’t allow them to be added to paints and varnishes as a pigment. Polyaniline (PANi), the product of aniline polymerization in the presence of various oxidizing agents, is an exception. The facility to cover the substrate and its poor solubility in almost all dispersed phases makes it possible to use it as a pigment in polymer composite materials”
It is suggested that the authors highlight exceptional properties of PANI in the light of latest literature ( for guidance please read Colloids and Surfaces A: Physicochemical and Engineering Aspects 626 (2021) 127076, Electrochimica Acta 320 (2019) 1345, Materials 2019, 12, 1626; Nanomaterials 2020, 10, 118 and others).
Polyaniline has been researched extensively and there are reports on the synthesis of soluble PANI. Authors should discuss it in more detail (please read Materials 12, 1626 (2019), Materials 12, 1227 (2019) and Synth. Metals 162 (2012) 2259-2266.
Response 3: The articles that have been recommended by the reviewer were also analyzed as well as the other sources. The Introduction section of the manuscript has been revised based on this data.
Point 4 (Materials and methods): Incorporate reference for equation 3.
Response 4: The reference to the equation (3) was added to the manuscript.
Point 5 (Results and Discussion): It would be more appropriate to characterize PANI shells with some more spectroscopic techniques such as FTIR.
Response 5: The results of the study obtained using IR spectroscopy, optical microscopy and SEM were also added to the manuscript.
Point 6 (Results and Discussion): The better performance of doped PANI shell has been attributed to the large surface area of PANI. However, one cannot find surface area measurements in the whole manuscript. Similarly morphological characterization would add additional merits to the PANI based shells for paints and varnishes composition development.
Response 6: Unfortunately, studies to determine the specific surface area are currently underway, but not completed.
Point 7 (Results and Discussion): Is there any correlation between the electrical conductivity and physicotechnical pigment properties of PANI based shell?
Response 7: Such regularities were not established in this work. Most likely, the optical characteristics will increase with the increase in the conductivity content, as well as the electrical conductivity.
Point 8 (References): Out of the 30 references, 3 are from the corresponding author of this article. This reflects self-citation.
Response 8: A total number of references has been increased and the number of citations of our own works had been reduced in the manuscript.
Round 2
Reviewer 1 Report
The revised manuscript is improved according to the reviewer's comments, which is suitable to be accepted for publication.
Author Response
Point 1: The revised manuscript is improved according to the reviewer's comments, which is suitable to be accepted for publication.
Response 1: We would like to express our deepest thanks for your remarks and comments; they helped us to improve our manuscript.
Reviewer 2 Report
The authors have provided revised version of their manuscript. Although they have complied with most of the earlier comments, there is still room for further improvement in the light of following comments.
- The authors write “The absorption band at 1478 cm-1 and 1560 cm-1 are assigned to benzenoid and quinoid ring vibration [38]. The absorption band characteristic of the conducting protonated form is found at about 1245 cm-1 . The absorption band at 1139 cm-1 is due to the in-plane bending vibration of C–H [39], while the 1305 cm-1 is because of C–N stretching of the benzenoid ring of PANI [40]. The aromatic-ring and out-of-plane C–H deformation vibrations manifest themselves in the region of 900–700 cm-1. The FTIR spectra of core/shell pigments are shown a blue shift of the main absorption bands compared to PANI”
One cannot find the figure relating to FTIR in the manuscript. FTIR figure number should also be cited in the text.
The authors should also analyze other bands ( if present) arising from Kaolin, Bentonite,Talc and Muscovite in the FTIR spectra of core/shell pigments.
- The authors have complied to one of my earlier comments “The better performance of doped PANI shell has been attributed to the large surface area of PANI. However, one cannot find surface area measurements in the whole manuscript” that “Unfortunately, studies to determine the specific surface area are currently underway, but not completed”
They need to incorporate the data on surface area measurements in the final draft of manuscript. Alternately they need to revise the relevant text in the manuscript to avoid exaggeration.
Author Response
Point 1:
- The authors write “The absorption band at 1478 cm-1 and 1560 cm-1 are assigned to benzenoid and quinoid ring vibration [38]. The absorption band characteristic of the conducting protonated form is found at about 1245 cm-1 . The absorption band at 1139 cm-1 is due to the in-plane bending vibration of C–H [39], while the 1305 cm-1 is because of C–N stretching of the benzenoid ring of PANI [40]. The aromatic-ring and out-of-plane C–H deformation vibrations manifest themselves in the region of 900–700 cm-1. The FTIR spectra of core/shell pigments are shown a blue shift of the main absorption bands compared to PANI”
One cannot find the figure relating to FTIR in the manuscript. FTIR figure number should also be cited in the text.
The authors should also analyze other bands ( if present) arising from Kaolin, Bentonite,Talc and Muscovite in the FTIR spectra of core/shell pigments.
.
Response 1: Spectra are presented in Supplementary Information. The reference to the spectra is added to the manuscript. The commentary about the typical absorption bands of the fillers that were used as a core is also added to the manuscript.
Point 2: The authors have complied to one of my earlier comments “The better performance of doped PANI shell has been attributed to the large surface area of PANI. However, one cannot find surface area measurements in the whole manuscript” that “Unfortunately, studies to determine the specific surface area are currently underway, but not completed”
They need to incorporate the data on surface area measurements in the final draft of manuscript. Alternately they need to revise the relevant text in the manuscript to avoid exaggeration.
Response 2: The changes were added to the manuscript.